# The association between platelet-to-albumin ratio and diabetic peripheral neuropathy: A cross-sectional study in the Chinese population

Wenting Deng[1,2,3,4,5,6], Siqi Zhang[6], Yueyang Zhang[1,2,3,4,5,6], Qin Wan🆔[1,2,3,4,5]*

1 Department of Endocrinology and Metabolism, The Affiliated Hospital of Southwest Medical University, Luzhou, China, 2 Metabolic Vascular Disease Key Laboratory of Sichuan Province, Luzhou, China, 3 Sichuan Clinical Research Center for Diabetes and Metabolism, Luzhou, China, 4 Sichuan Clinical Research Center for Nephropathy, Luzhou, China, 5 Cardiovascular and Metabolic Diseases Key Laboratory of Luzhou, Luzhou, China, 6 Southwest Medical University, Luzhou, China

* wanqin360@swmu.edu.cn

## Abstract

### Introduction

The early prevention and diagnosis of diabetic complications are challenging, particularly for diabetic peripheral neuropathy (DPN), which progresses insidiously and irreversibly. Currently, reliable biomarkers associated with DPN are still limited. This study explored the association between the platelet-to-albumin ratio (PAR) and diabetic peripheral neuropathy in Chinese adults with type 2 diabetes mellitus(T2DM).

### Methods

This cross-sectional study included adult patients with T2DM enrolled in the metabolic management center (MMC) database at the Affiliated Hospital of Southwest Medical University between June 2018 and December 2023. Participants meeting the predefined eligibility criteria were classified into diabetic peripheral neuropathy (DPN) and non-diabetic peripheral neuropathy (non-DPN) groups. The association between PAR and DPN was evaluated using logistic regression models, with subgroup analyses and restricted cubic spline regression used to examine potential effect modification and dose–response relationships. Receiver operating characteristic (ROC) curve analysis was performed to assess the discriminative ability of PAR for identifying DPN.

### Results

Among 1,141 patients with T2DM, including 427 patients with DPN, PAR levels were significantly higher in the DPN group than in the non-DPN group ($P < 0.001$). Univariate analysis showed that higher PAR (OR=1.282, 95% CI: 1.183–1.389, $P < 0.001$)

**Data availability statement:** The data used in this study were obtained from the National Metabolic Management Centre (MMC), a nationwide, registered clinical management system in China. The dataset contains sensitive clinical data and potentially identifiable patient information collected under ethical approval and specific informed consent for clinical management and research purposes. Therefore, the data are not publicly available due to privacy and ethical restrictions imposed by the MMC data governance policy. De-identified data may be made available from the corresponding author upon reasonable request and subject to approval by the MMC data governance committee.

**Funding:** This study was supported by the General Program of Science and Technology Department of Sichuan Province in the form of a grant awarded to QW (2024NSFSC0594), by the Ministry of Science and Technology of China in the form of a grant awarded to QW (2016YFC0901200), and by Southwest Medical University in the form of a grant awarded to QW (2019SQN013). The funders had no role in study design, data collection and analysis, decision to publish, or preparation of the manuscript.

**Competing interests:** The authors have declared that no competing interests exist.

**Abbreviations:** T2DM, type 2 diabetes mellitus; DPN, diabetic peripheral neuropathy; MMC, National Metabolic Management Centre; ADA, the American Diabetes Association; OGTT, oral glucose tolerance test; DBP, diastolic blood pressure; SBP, systolic blood pressure; BMI, body mass index; FPG, fasting plasma glucose; 2hPG, postprandial 2-hour plasma glucose; HbA1c, hemoglobin A1c; Hb, hemoglobin; RBC, red blood cell; WBC, white blood cell; PLT, platelet; MPV, mean platelet volume; HCT, hematocrit; MCV, mean erythrocyte volume; MCH, red cell hemoglobin volume; MPV, mean platelet volume; ALT, alanine aminotransferase; AST, aspartate aminotransferase; ALB, albumin; BUN, blood urea nitrogen; Cr, creatinine; eGFR, estimated glomerular filtration rate; UA, uric acid; TG, triglycerides; TC, total cholesterol; HDL-C, high-density lipoprotein cholesterol; LDL-C, low-density lipoprotein cholesterol;

significantly increased the prevalence of DPN. After adjusting for confounders, the platelet-to-albumin ratio (PAR) remained independently associated with DPN (OR = 1.158, 95% CI: 1.0511.276, $P = 0.003$). In subgroup analysis, no significant interaction was observed (all $P$ for interaction > 0.05). The restricted cubic spline regression confirmed a positive linear association between PAR levels and DPN ($P = 0.001$, P for nonlinearity = 0.986). Additionally, ROC analysis identified an optimal PAR cut-off value of 5.5971 for distinguishing patients with and without DPN.

## Conclusions

In patients with T2DM, higher platelet-to-albumin ratio(PAR) levels were associated with the prevalence of diabetic peripheral neuropathy. This association may be related to underlying inflammatory processes involved in DPN.

## 1. Introduction

The global prevalence of diabetes is increasing significantly due to changing lifestyles and extended life expectancies. Currently, approximately 537 million individuals worldwide live with diabetes, and this figure is projected to surge to 1.3 billion by 2050 [1]. In China, 140 million individuals were diagnosed with diabetes in 2021, and over 90% of these patients were classified as type 2 diabetes mellitus (T2DM) [2]. Diabetic peripheral neuropathy (DPN) is a common complication in T2DM; approximately 50% of diabetic patients progress to diabetic peripheral neuropathy as the duration of their diabetes increases [3]. DPN is characterized by symptoms and signs of peripheral nerve dysfunction [4]. It predominantly manifests as distal symmetric polyneuropathy, which includes symptoms such as bilateral limb numbness, tingling sensations, burning pain, electrocution-like pain, and various sensory abnormalities [5]. The incidence of DPN is typically insidious, and by the time it is diagnosed, significant nerve damage may have already occurred, often rendering it irreversible [6]. However, effective approaches for the early detection of DPN remain limited. Therefore, identifying biological indicators associated with DPN may facilitate earlier recognition and improved clinical management,which could ultimately contribute to better patient outcomes and quality of life.

Inflammatory processes play a pivotal role in the onset and progression of T2DM and its complications. Several inflammatory markers, such as C-reactive protein (CRP), interleukins (ILs), and tumor necrosis factor-alpha (TNF-α), have been associated with an increased risk of diabetic complications [7,8]. For example, previous studies have shown that nuclear factor kappa B (NF-κB), a key transcription factor involved in inflammatory responses, contributes to endothelial dysfunction by regulating the expression of inflammatory genes, thereby promoting Schwann cell apoptosis and myelin sheath degradation, which further exacerbates neuronal damage [9]. Evidence indicates that low levels of systemic inflammation are associated with DPN [10]. Moreover, the platelet to serum albumin ratio (PAR) is an emerging indicator of systemic inflammation that has received significant attention in recent

studies [11–13]. Accumulating evidence has demonstrated a robust association between PAR and several inflammatory and thrombotic diseases, including diabetic nephropathy, ankylosing spondylitis, IgA nephropathy, cardiovascular disease, neoplasms, and adverse peritoneal dialysis outcomes [14–21]. However, no research has specifically examined the relationship between PAR and DPN. Therefore, this study aimed to investigate the association between platelet-to-albumin ratio (PAR), a composite marker reflecting inflammatory and nutritional status, and the prevalence of diabetic peripheral neuropathy among patients with T2DM in a Chinese population.

## 2. Materials and methods

### 2.1. Study population

The study was a cross-sectional analysis based on data obtained from the National Metabolic Management Centre (MMC) program at the Affiliated Hospital of Southwest Medical University, a participating center in the nationwide MMC network in China. The MMC is a standardized metabolic disease management program that collects routine clinical data using unified protocols, and all data used in the present study were de-identified before analysis [22–24]. Patients in the MMC program receive continuous follow-up as part of routine clinical management. For this cross-sectional analysis, only clinical and laboratory data obtained at the first MMC visit were used. Data extraction was performed in December 2023 from the MMC database. Data were collected by trained endocrinology physicians and nurses at participating centers following standardized operating procedures, and all personnel received uniform training on study-specific questionnaires and outcome measures before data collection. Each participant underwent a comprehensive evaluation that included standardized questionnaires, a thorough physical examination, and a battery of laboratory tests. The participant selection process is shown in Fig 1. Eligible participants were adults (age ≥ 18 years) with confirmed T2DM according to the American Diabetes Association (ADA) "Standards of Care in Diabetes" [25]; Exclusion criteria:1) type 1 diabetes, gestational diabetes and other special types of diabetes; 2)Use of oral antiplatelet and anticoagulant drugs;3)History of infection within 3 months, severe diabetic foot complications, or severe liver or kidney disease; 4)Patients with malignant tumors, hematological disorders; 5)Missing data. Diabetes duration was determined based on patient-reported time since diagnosis, collected during standardized clinical visits and follow-up.

### 2.2. Ethics approval

All participants provided written informed consent at the time of enrollment in the Metabolic Management Center (MMC) program, which included permission for the use of their anonymized clinical data for research purposes. The study was conducted according to the ethical guidelines of the 2024 Declaration of Helsinki (latest revision) and was approved by the ethics committee of the Affiliated Hospital of Southwest Medical University (ethical approval code: 2018017, date: February 2018).

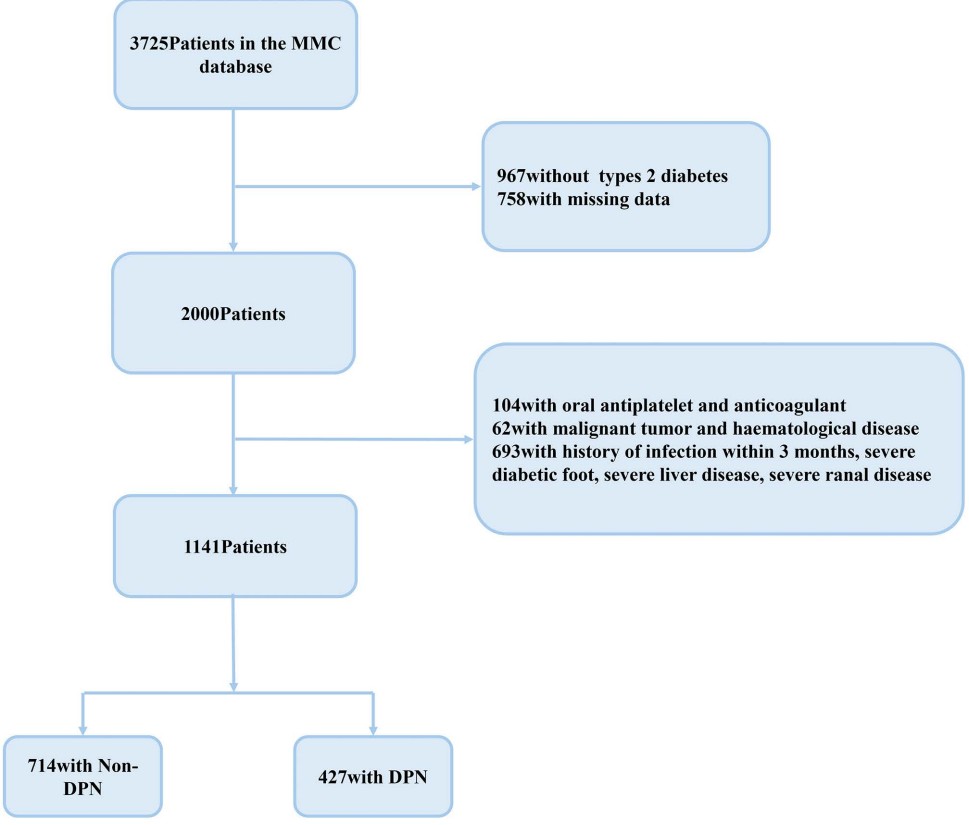

**Fig 1. Flowchart for the selection of the participants from the MMC database.** This flowchart illustrates the inclusion and exclusion process of participants with type 2 diabetes mellitus (T2DM) in the present cross-sectional study. A total of 1,141 eligible subjects were enrolled and categorized into diabetic peripheral neuropathy (DPN) and non-DPN groups based on clinical and electrophysiological criteria.

## 2.3. Diagnostic criteria

T2DM was defined according to the American Diabetes Association "Standards of Care in Diabetes" [25]: the fasting plasma glucose (FPG) ≥7.0 mmol/L, and/or two-hour plasma glucose (2h PG) value during a 75g oral glucose tolerance test (OGTT) ≥ 11.1 mmol/L or hemoglobin A1c (HbA1c) ≥ 6.5% or self-reported medical history.

Diagnosis of diabetic peripheral neuropathy was based on standardized clinical criteria. All assessments were performed by trained endocrinologists or certified technicians who had received uniform training in neuropathy evaluation before the study and who followed standardized operating procedures across participating centers. Patients were required to have a confirmed diagnosis of type 2 diabetes mellitus, with neuropathic symptoms occurring at or after diagnosis of diabetes. Neurological symptoms, including numbness, pain (tingling or prickling, pricking, burning, or aching), and sensory abnormalities (abnormal cold or heat sensations, nociceptive hypersensitivity, and dysesthesia) in the toes, feet, legs, or upper limbs, were assessed by structured questioning. Bilateral Achilles tendon reflexes were examined in the knee standing position and recorded as present, diminished, or absent [26]. The vibration perception threshold (VPT) was assessed at the metatarsophalangeal joint of the hallux using a neurothesiometer (Bio-Thesiometer; Bio-Medical Instrument Co., Newbury, OH, USA). The patients were first instructed on how to recognize the vibration sensation. The stimulus amplitude was gradually increased from zero, and the patients were asked to indicate when the vibration was first perceived. Measurements were performed three times on the plantar surface of each hallux, and the

median value was recorded as the VPT for each foot. Sensitivity to touch was evaluated using a 5.07/10-g Semmes-Weinstein monofilament (SWM) at four standardized sites on each foot(three plantar sites and one dorsal side). The monofilament was applied perpendicular to the skin until it just buckled, and was maintained for approximately 2 seconds. Diabetic peripheral neuropathy was defined as the VPT ≥ 25 V and/or inability to perceive the 10-g monofilament at one or more test sites [27].

## 2.4. Data collection

Clinical data were collected as follows: The patient's demographic information includes sex and age. The questionnaire collected patient information about the history of hypertension, diabetes mellitus, hyperlipidemia, smoking, alcohol consumption, and medication. The patients' height and weight were also measured. Blood pressure was measured in the morning after at least 5 minutes of rest. Systolic blood pressure (SBP), diastolic blood pressure (DBP), and heart rate were measured on the right arm using a standard mercury sphygmomanometer, with three readings taken and the average calculated. FPG and 2h PG were measured using venous blood samples collected during a standard OGTT, with samples obtained after an overnight fast and again 2 hours after glucose ingestion. Additional morning fasting blood samples were collected from each patient after an 8-hour fast. These samples were used to measure alanine aminotransferase (ALT), aspartate aminotransferase(AST), plasma albumin(ALB), Alkaline phosphatase (ALP), gamma-glutamyl transpeptidase (γ-GGT), blood urea nitrogen(BUN), uric acid(UA), creatinine(Cr), glycated hemoglobin A1c (HbA1c), White blood cells (WBC), red blood cells (RBC), mean platelet volume (MPV), Platelet(PLT), Hematocrit (HCT), Mean Corpuscular Volume (MCV), Mean Corpuscular Hemoglobin (MCH), low-density lipoprotein cholesterol (LDL-C), high-density lipoprotein cholesterol (HDL-C), total cholesterol (TC) and triglycerides (TG), were determined according to established protocols and guidelines at the registered central laboratory of the Affiliated Hospital of Southwestern Medical University, which is accredited in line with the international organization for standardization (ISO) 15189 standard for quality management specific to medical laboratories.

Urinary albumin and creatinine were measured from the first urine sample. The urinary microalbumin and creatinine concentrations were determined via chemiluminescent immunoassay, and the urinary albumin-to-creatinine ratio (ACR) was calculated and reported in milligrams per gram (mg/g).

## 2.5. Calculations

Body mass index (BMI) was calculated as weight (kg) divided by height squared (m²). The platelet to serum albumin ratio (PAR) was calculated as platelet count (10^9/L) divided by serum albumin concentration (g/L). The platelet-to-high-density lipoprotein cholesterol ratio (PHR) was calculated as platelet count (10^9/L) divided by HDL-C (mmol/L). Estimated glomerular filtration rate (eGFR) was calculated using the Modification of Diet in Renal Disease (MDRD) equation, incorporating serum creatinine concentration, age, and sex, as previously described.

$$BMI = Weight(kg)/[Height(m)^2]$$

$$eGFR = 170 \times [Cr(mg/dL)]^{-1.234} \times (Age)^{-0.179} \times (0.79 \text{ if female})$$

$$PAR = PLT(10^9/L)/ALB(g/L)$$

$$PHR = PLT(10^9/L)/HDL-C(mmol/L)$$

## 2.6. Statistical analysis

The statistical analyses were conducted using SPSS 27.0, and restricted cubic spline regression (RCS) and subgroup analyses were performed using RStudio. Participant characteristics were presented as either mean (standard deviation) or median (interquartile range), depending on the distribution of continuous variables. Continuous variables with normal distribution were analyzed using Student's t-test, while non-normally distributed variables were assessed through the Mann-Whitney U test. Categorical variables were expressed as counts (proportions). Comparisons between groups were made using the Chi-square test or Fisher's exact test. Patients with T2DM were classified into four groups according to PAR quartiles. Logistic regression models were used to evaluate the association between PAR and DPN. The analysis was adjusted for several factors, including age, sex, smoking, alcohol consumption, duration of diabetes, SBP, BMI, FPG, 2hPG, glycated hemoglobin (HbA1c), Heart rate, WBC, MPV, ALT, AST, ALP, γ-GGT, BUN, TG, HDL-C, LDL-C, urinary ACR, and eGFR; covariates were selected based on clinical relevance, prior literature, and univariate analyses. Multicollinearity was assessed using variance inflation factors (VIF) derived from a linear regression model including all candidate predictors; redundant covariates with VIF > 5 were excluded. Variables that represent clinical manifestations or consequences of diabetic peripheral neuropathy, such as a history of diabetic foot or nerve conduction study results, were not included as covariates, as they are downstream outcomes of neuropathy rather than independent confounding factors. Subgroup analyses were performed to investigate potential effect modification, and interactions were examined by adding interaction terms to the regression models. To further assess the association between PAR and DPN across its continuous distribution, RCS models with four knots were fitted. Receiver Operating Characteristic (ROC) curves assessed the discriminative ability of PAR for identifying DPN. All statistical analyses were conducted with two-tailed p-values, with significance set at $p < 0.05$.

## 3. Results

### 3.1. Clinical and laboratory characteristics in DPN patients

The study included 1141 patients diagnosed with T2DM, divided into two subgroups based on the presence or absence of diabetic peripheral neuropathy (DPN). The group with non-DPN comprised 714 patients, while the DPN group included 427 patients. A detailed overview of the clinical and laboratory data of the study population is shown in **Table 1**. When compared with the non-DPN group, the DPN group exhibited significantly higher DBP, SBP, heart rate, FPG, 2hPG, HbA1C, WBC, PLT, BUN, TC, LDL-C, albumin-to-creatinine ratio (ACR), PAR, and PHR($P < 0.05$). In contrast, the DPN group had a lower average age, weight, BMI, Hb, RBC, HCT, MCV, MCH, ALT, AST, ALB, history of hypertensive and history of antihypertensive drugs($P < 0.05$) compared to the non-DPN group.

### 3.2. Univariate and multivariate analysis of determinants of DPN in T2DM patients

**Table 2** shows the results of univariable and multivariable logistic regression analyses of factors associated with DPN. In the univariable analysis, History of antihypertensive drugs, age, duration of diabetes, DBP, SBP, heart rate, BMI, HbA1c, Hb, RBC, WBC, HCT, MCV, MCH, ALT, AST, BUN, Cr, TC, LDL-C, log(ACR), and PAR were significantly associated with DPN (P < 0.05). In the multivariable model, age, history of antihypertensive drugs, duration of diabetes, SBP, heart rate, BMI, HbA1c, BUN, log(ACR), and PAR were significantly and independently associated with DPN ($P < 0.05$). Notably, higher PAR remained significantly associated with higher odds of DPN after adjustment for multiple potential confounders(OR=1.158,95% CI, 1.051,1.276, $P < 0.05$).

### 3.3. Association of PAR quartiles with prevalent DPN in T2DM patients

**Table 3** shows that PAR was treated as a categorical variable and analyzed using binary logistic regression. Compared with the lowest quartile(T1), the odds of DPN increased across higher PAR quartiles. After adjusting for age, sex, smoking,

**Table 1. Clinical and laboratory characteristics in patients with T2DM.**

| Variables | Non-DPN(n = 714) | DPN(n = 427) | P value |
|---|---|---|---|
| Sex(male) | 396(55.46%) | 238(55.74%) | 0.928 |
| Age(years) | 56.21(50.00, 65.00) | 54.41(48.00, 64.00) | 0.007* |
| Duration of diabetes (months) | 76.95(5.00, 125.00) | 86.65(11.00, 140.00) | 0.14 |
| DBP (mmHg) | 78.11(70.00, 85.00) | 80.92(73.00, 88.00) | 0.001* |
| SBP (mmHg) | 132.57(120.00, 144.00) | 137.34(120.00, 152.00) | 0.004* |
| Heart rate | 73.64(65.00, 80.00) | 78.05(69.00, 86.00) | <0.001* |
| BMI (kg/m²) | 25.06(22.80, 26.90) | 24.13(21.80, 26.30) | <0.001* |
| FPG (mmol/L) | 9.46(7.30, 11.50) | 10.37(7.80, 12.40) | <0.001* |
| 2hPG(mmol/L) | 14.38(10.80, 17.50) | 15.39(12.10, 18.70) | <0.001* |
| HbA1c (%) | 9.37(7.50, 10.80) | 10.54(8.80, 12.20) | <0.001* |
| Hb(g/L) | 140.72(130.00, 153.00) | 136(124.00,149.00) | <0.001* |
| RBC (10^12/L) | 4.67(4.30, 5.04) | 4.57(4.13, 4.98) | 0.009* |
| WBC (10^9/L) | 6.61(5.34, 7.50) | 6.96(5.51, 7.89) | 0.017* |
| PLT (10^9/L) | 202.27(164.00, 234.00) | 220.22(175.00,255.00) | <0.001* |
| HCT (%) | 0.42(0.4, 0.46) | 0.41(0.38, 0.44) | <0.001* |
| MCV (fl) | 91.1(88.90, 94.50) | 89.9(87.50, 93.30) | <0.001* |
| MCH (pg) | 30.21(29.40,31.40) | 29.86(29.00,31.30) | 0.005* |
| MPV (fl) | 11.24(10.20, 12.20) | 11.19(10.20,12.10) | 0.595 |
| ALT (U/L) | 31.54(16.60,35.60) | 26.33(14.60,30.20) | <0.001* |
| AST (U/L) | 24.73(16.30,26.70) | 21.96(14.60,23.70) | <0.001* |
| ALP (U/L) | 86.19(65.70,100.10) | 88.11(65.90,102.60) | 0.336 |
| γ-GGT(U/L) | 26.5(17.70,44.90) | 24.8(16.30,42.90) | 0.137 |
| ALB (g/L) | 44.63(42.70,47.00) | 43.3(40.70,46.40) | <0.001* |
| BUN (mmol/L) | 5.79(4.72,6.56) | 6.24(4.67,7.31) | 0.001* |
| Cr (µmmol/L) | 62.91(50.00,71.40) | 66.07(48.30,75.50) | 0.455 |
| eGFR (mL/min/1.73 m²) | 125.71(101.81,148.02) | 128.3(90.2,157.89) | 0.948 |
| UA(µmmol/L) | 340.72(267.70,388.70) | 332.01(261.50,392.30) | 0.891 |
| TG(mmol/L) | 2.56(1.23,2.81) | 2.47(1.22,2.81) | 0.736 |
| TC(mmol/L) | 4.7(3.87,5.40) | 4.96(4.04,5.54) | 0.038* |
| HDL-c(mmol/L) | 1.15(0.92,1.29) | 1.18(0.92,1.34) | 0.221 |
| LDL-c(mmol/L) | 2.8(2.07,3.45) | 2.96(2.23,3.61) | 0.034* |
| Urinary ACR (mg/g) | 13.15(7.10,34.60) | 24.00(9.90,108.20) | <0.001* |
| PAR | 4.56(3.63,5.27) | 5.17(3.98,5.98) | <0.001* |
| PHR | 187.98(136.91,225.42) | 203.75(142.73,243.68) | 0.017* |
| Smoking | | | 0.382 |
| Never | 444(62.18%) | 259(60.66%) | |
| Ever | 46(6.44%) | 21(4.92%) | |
| Current | 224(31.37%) | 147(34.43%) | |
| Alcohol consumption | | | 0.406 |
| Never | 369(51.68%) | 236(55.27%) | |
| Ever | 85(11.9%) | 52(12.18%) | |
| Current | 260(36.41%) | 139(32.55%) | |
| history of hypertension | | | 0.005* |
| No | 447(62.61%) | 302(70.73%) | |

*(Continued)*

**Table 1.** (Continued)

| Variables | Non-DPN(n = 714) | DPN(n = 427) | P value |
|---|---|---|---|
| Yes | 267(37.39%) | 125(29.27%) | |
| history of hyperlipidemia | | | 0.141 |
| No | 601(84.17%) | 373(87.35%) | |
| Yes | 113(15.83%) | 54(12.65%) | |
| History of high uric acid | | | 0.191 |
| No | 677(94.82%) | 412(96.49%) | |
| Yes | 37(5.18%) | 15(3.51%) | |
| History of insulin | | | 0.2 |
| No | 295(41.32%) | 193(45.20%) | |
| Yes | 419(58.68%) | 234(54.80%) | |
| History of antihypertensive drugs | | | <0.001* |
| No | 526(73.67%) | 368(86.18%) | |
| Yes | 188(26.33%) | 59(13.82%) | |

Data are median (interquartile range) for continuous variables or n (percentage) for categorical variables. DPN, diabetic peripheral neuropathy; DBP, diastolic blood pressure; SBP, systolic blood pressure; BMI, body mass index; FPG, fasting blood glucose; 2hPG, postprandial 2-hour plasma glucose; HbA1c, hemoglobin A1c; Hb, hemoglobin; RBC, red blood cell; WBC, white blood cell; PLT, platelet; MPV, mean platelet volume; HCT, hematocrit; MCV, mean corpuscular volume; MCH, mean corpuscular hemoglobin; ALT, alanine aminotransferase; AST, aspartate aminotransferase; ALB, albumin; BUN, blood urea nitrogen; Cr, creatinine; eGFR, estimated glomerular filtration rate; UA, uric acid; TG, triglycerides; TC, total cholesterol; HDL-C, high-density lipoprotein cholesterol; LDL-C, low-density lipoprotein cholesterol; urinary ACR, urinary albumin- to-creatinine ratio; PAR, platelet-to-albumin ratio; PHR, platelet-to- high-density lipoprotein cholesterol ratio; *p<0.05.

alcohol consumption, duration of diabetes, SBP, BMI, FPG, 2hPG, HbA1c, Heart rate, WBC, MPV, ALT, AST, ALP, γ-GGT, BUN, TG, HDL-C, LDL-C, log(ACR), eGFR, only the highest quartiles(T4) remained significantly associated with DPN(OR =2.093, $P<0.05$), whereas the associations for T2 and T3 were directionally positive but did not reach statistical significance. These findings suggest that the association between PAR and DPN became statistically more evident at higher PAR levels.

### 3.4. Subgroup analysis of the associations between PAR and DPN

The results of the subgroup analysis suggested that the association between PAR and DPN was not uniform across strata, as illustrated in Fig 2. When stratified by age, sex, body mass index (BMI), duration of diabetes, history of hypertension, HbA1c, and HDL-C levels, the highest PAR quartile (T4) was generally associated with higher odds of DPN compared with the lowest quartile (T1); however, this association did not reach statistical significance in all subgroups (S1 Fig). Significant associations for T4 were observed in several strata, whereas in others the confidence intervals crossed unity. Nevertheless, interaction testing indicated that none of the examined subgroup variables significantly modified the association between PAR and DPN (all P for interaction > 0.05).

### 3.5. Restricted cubic spline and ROC analyses of PAR in relation to prevalent DPN

The restricted cubic spline regression(RCS) analysis showed an overall positive linear association between the PAR and DPN, both before and after adjustments for covariates (unadjusted in Model 1: $P<0.001$, P for nonlinearity = 0.708; Adjusted covariates in Model 3: $P=0.001$, P for nonlinearity = 0.986). Fig 3A-3C). To explore the discriminative ability of PAR for DPN, ROC analysis was then performed to evaluate the discriminative ability of ALB, PLT, PAR, and PHR for distinguishing patients with and without DPN (Fig 3(D)). The results indicated that, compared with ALB, PLT, and PHR, PAR exhibited the highest discriminative performance for DPN, with an area under the curve (AUC) of 0.602 (95% CI: 0.568, 0.636; P<0.001).

**Table 2. Univariate and multivariate logistic analysis of factors associated with DPN.**

| Variables | Univariate Analysis | | | Multivariate Analysis | | |
|---|---|---|---|---|---|---|
| | B | OR (95% CI) | P value | B | OR (95% CI) | P value |
| Sex (male) | −0.011 | 0.989(0.777,1.259) | 0.928 | | | |
| Age | −0.015 | 0.986(0.975,0.996) | 0.008* | −0.021 | 0.979(0.965,0.993) | 0.002* |
| Smoking | | | 0.383 | | | |
| Never | − | − | | | | |
| Ever | −0.245 | 0.783(0.457,1.341) | 0.372 | | | |
| Current | 0.118 | 1.125(0.869,1.457) | 0.372 | | | |
| Alcohol consumption | | | 0.406 | | | |
| Never | − | − | | | | |
| Ever | −0.044 | 0.957(0.653,1.401) | 0.820 | | | |
| Current | −0.179 | 0.836(0.643,1.087) | 0.181 | | | |
| history of hyperlipidemia | −0.261 | 0.77(0.543,1.092) | 0.142 | | | |
| History of insulin | −0.158 | 0.854(0.67,1.087) | 0.200 | | | |
| History of antihypertensive drugs | −0.802 | 0.449(0.325,0.619) | <0.001* | −0.922 | 0.398(0.271,0.584) | <0.001* |
| Duration of diabetes | 0.002 | 1.002(1,1.003) | 0.036* | 0.003 | 1.003(1.001,1.005) | 0.002* |
| DBP | 0.022 | 1.022(1.011,1.033) | <0.001* | | | |
| SBP | 0.012 | 1.012(1.006,1.018) | <0.001* | 0.010 | 1.010(1.002,1.018) | 0.015 |
| Heart rate | 0.027 | 1.027(1.017,1.037) | <0.001* | 0.022 | 1.023(1.012,1.034) | <0.001* |
| BMI | −0.075 | 0.928(0.895,0.961) | <0.001* | −0.066 | 0.936(0.899,0.975) | 0.001* |
| HbA1c% | 0.199 | 1.22(1.159,1.284) | <0.001* | 0.204 | 1.226(1.154,1.302) | <0.001* |
| Hb | −0.014 | 0.986(0.979,0.993) | <0.001* | | | |
| RBC | −0.259 | 0.772(0.635,0.939) | 0.009* | | | |
| WBC | 0.081 | 1.084(1.022,1.15) | 0.007* | −0.011 | 0.990(0.923,1.061) | 0.771 |
| HCT | −5.770 | 0.003(0,0.034) | <0.001* | | | |
| MCV | −0.028 | 0.972(0.954,0.991) | 0.003* | −0.008 | 0.992(0.972,1.013) | 0.441 |
| MCH | −0.064 | 0.938(0.891,0.988) | 0.015* | | | |
| MPV | −0.024 | 0.977(0.902,1.058) | 0.560 | | | |
| ALT | −0.009 | 0.991(0.985,0.997) | 0.002* | −0.006 | 0.994(0.984,1.003) | 0.167 |
| AST | −0.012 | 0.988(0.979,0.997) | 0.011* | 0.005 | 1.005(0.992,1.018) | 0.466 |
| ALP | −0.002 | 1.002(0.998,1.005) | 0.335 | | | |
| γ-GGT | 0.0003 | 1(0.999,1.001) | 0.585 | | | |
| BUN | 0.127 | 1.136(1.065,1.211) | <0.001* | 0.097 | 1.102(1.020,1.191) | 0.014* |
| Cr | 0.006 | 1.006(1.001,1.011) | 0.024* | | | |
| eGFR | 0.001 | 1.001(0.999,1.004) | 0.342 | | | |
| UA | −0.001 | 0.999(0.998,1) | 0.274 | | | |
| TG | −0.015 | 0.985(0.938,1.034) | 0.547 | | | |
| TC | 0.139 | 1.149(1.048,1.259) | 0.003* | | | |
| HDL-C | 0.268 | 1.308(0.935,1.829) | 0.117 | | | |
| LDL-C | 0.155 | 1.167(1.037,1.314) | 0.010* | −0.031 | 0.969(0.846,1.110) | 0.650 |
| log(ACR) | 0.294 | 1.341(1.245,1.446) | <0.001* | 0.235 | 1.265(1.150,1.391) | <0.001* |
| PAR | 0.248 | 1.282(1.183,1.389) | <0.001* | 0.147 | 1.158(1.051,1.276) | 0.003* |

Notes: B represents the regression coefficient (log odds) and measures the influence of each variable on DPN. OR, odds ratio; CI, confidence intervals, *p<0.05.

**Table 3. Association of PAR quartiles with increased prevalence of DPN in stratified Analysis.**

| Variable | Model 1 | P value | Model 2 | P value | Model 3 | P value |
|---|---|---|---|---|---|---|
| PAR continue | 1.281(1.187,1.384) | <0.001 | 1.280(1.185,1.384) | <0.001 | 1.234(1.102,1.381) | <0.001 |
| PAR quartiles | | | | | | |
| T1 (<3.733) | 1 | | 1 | | 1 | |
| T2 (3.733–4.600) | 1.006(0.705,1.436) | 0.972 | 1.005(0.703,1.435) | 0.980 | 1.023(0.683,1.533) | 0.911 |
| T3 (4.600–5.608) | 1.218(0.857,1.73) | 0.271 | 1.19(0.835,1.696) | 0.337 | 1.131(0.731,1.749) | 0.580 |
| T4 (>5.608) | 2.638(1.872,3.718) | <0.001* | 2.617(1.848,3.706) | <0.001* | 2.093(1.302,3.364) | 0.002* |

Model 1: Unadjusted

Model 2: Adjusted for age and sex.

Model3: Adjusted for age, sex, smoking, alcohol consumption, duration of diabetes, SBP, BMI, FPG,2hPG, HbA1c, Heart rate, WBC, MPV, ALT, AST, ALP, γ-GGT, BUN, TG, HDL-C, LDL-C, urinary ACR, eGFR.

*p<0.05.

| Variable | | OR (95% CI) | P value | P for interaction |
|---|---|---|---|---|
| Overall | | 1.28 (1.18 to 1.39) | 0.001* | |
| Age | | | | 0.462 |
| <65 | | 1.3 (1.19 to 1.43) | 0.001* | |
| ≥65 | | 1.21 (1.03 to 1.43) | 0.022* | |
| Sex | | | | 0.504 |
| Male | | 1.25 (1.11 to 1.41) | 0.001* | |
| Female | | 1.33 (1.18 to 1.48) | 0.001* | |
| BMI | | | | 0.133 |
| <24 | | 1.37 (1.21 to 1.56) | 0.001* | |
| ≥24 | | 1.21 (1.09 to 1.35) | 0.001* | |
| Duration | | | | 0.843 |
| <5 | | 1.3 (1.09 to 1.56) | 0.004* | |
| ≥5 | | 1.28 (1.17 to 1.4) | 0.001* | |
| Hypertension history | | | | 0.41 |
| No | | 1.31 (1.19 to 1.45) | 0.001* | |
| Yes | | 1.22 (1.05 to 1.42) | 0.01* | |
| HAb1C | | | | 0.756 |
| <7 | | 1.21 (0.9 to 1.62) | 0.2 | |
| ≥7 | | 1.27 (1.17 to 1.38) | 0.001* | |
| HDL-c | | | | 0.077 |
| <1.04 | | 1.19 (1.06 to 1.33) | 0.003* | |
| ≥1.04 | | 1.37 (1.23 to 1.53) | 0.001* | |

0.5  1    2

**Fig 2. Subgroup analysis of the associations between PAR and DPN; *P<0.05.** Forest plots demonstrate the consistency of the association between PAR and DPN across predefined subgroups stratified by age, sex, BMI, duration of diabetes, hypertension history, HbA1c, and HDL-C levels. No significant interaction effects were observed (all P for interaction>0.05). Abbreviations: PAR, platelet-to-albumin ratio; DPN, diabetic peripheral neuropathy; BMI, body mass index; HbA1c, glycated hemoglobin; HDL-C, high-density lipoprotein cholesterol.

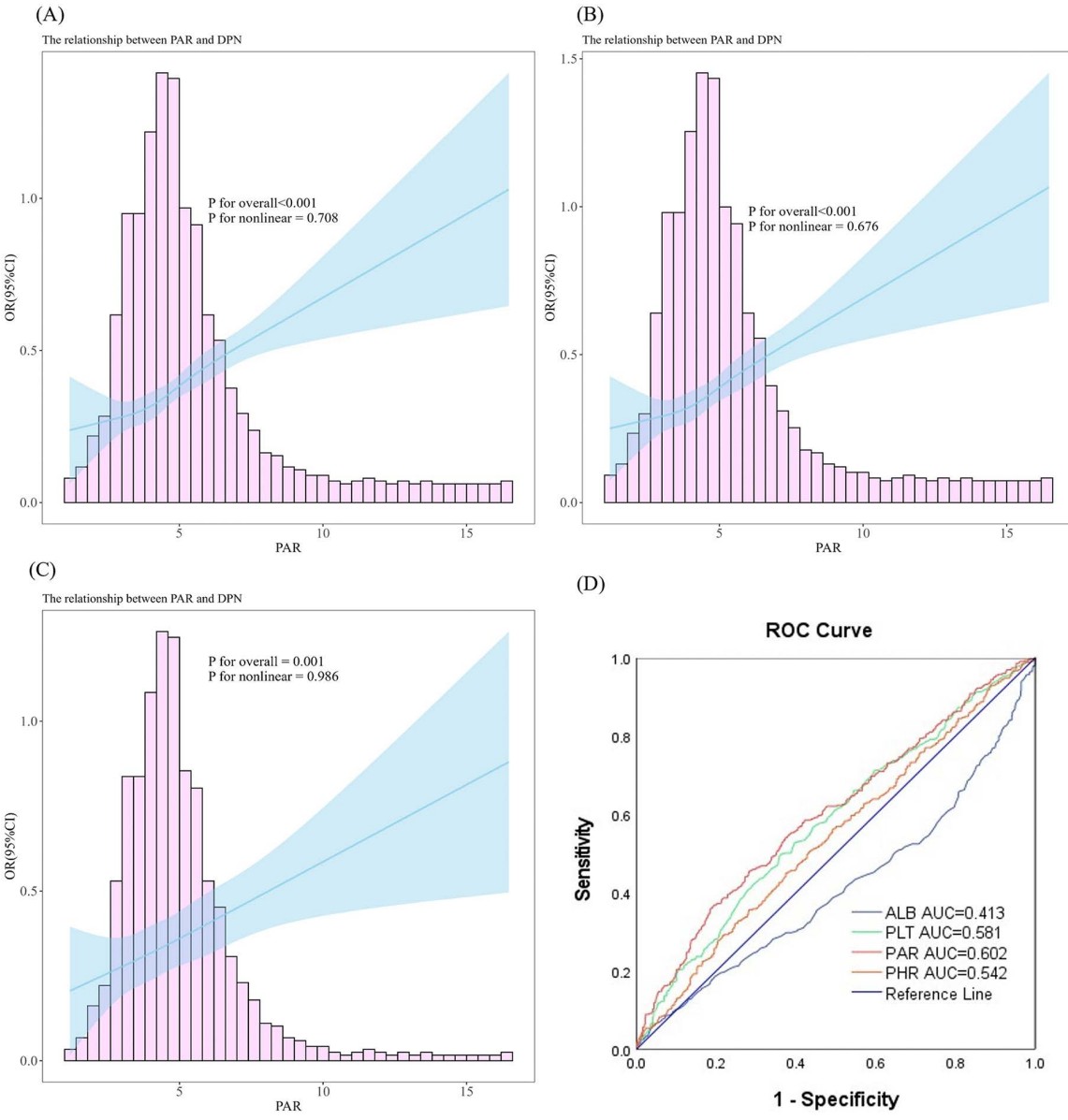

**Fig 3. Restricted cubic spline curves (RCS) and receiver operating characteristic (ROC) analysis of PAR in relation to prevalent DPN.** (A) Unadjusted RCS model. (B) RCS model adjusted for age and sex. (C) RCS model adjusted for age, sex, smoking, alcohol consumption, duration of diabetes, SBP, BMI, FPG,2hPG, HbA1c, Heart rate, WBC, MPV, ALT, AST, ALP, γ-GGT, BUN, TG, HDL-C, LDL-C, urinary ACR, eGFR. (D) ROC curves comparing the discriminative ability of PAR, PLT, ALB, and PHR for distinguishing patients with and without DPN. The RCS curves show an overall positive linear association between PAR and prevalent DPN. The solid line represents the adjusted odds ratio (OR), and the shaded area represents the 95% confidence interval (CI). The ROC curves compare the discriminative performance of four inflammatory markers—PAR, platelet count (PLT), serum albumin (ALB), and platelet-to-high-density lipoprotein ratio (PHR)—in identifying DPN among patients with T2DM. PAR demonstrated the highest discriminative performance (AUC = 0.602, 95% CI: 0.568–0.636, P < 0.001), with an optimal cut-off value of 5.5971 (sensitivity 35.9%, specificity 81.7%). The AUC values and corresponding 95% confidence intervals for each indicator are summarized in Supplementary S1 Table.

It was followed by PLT (AUC: 0.581, 95%CI: 0.546, 0.615, *P*<0.001), PHR (AUC: 0.542, 95%CI: 0.507, 0.577, *P*=0.017), and ALB (AUC: 0.413, 95%CI: 0.378, 0.448, *P*<0.001)(S1 Table). By calculating the Youdon index, we found that the optimal cutoff value for PAR is 5.5971. At this threshold, PAR demonstrated a sensitivity of 35.9% and a specificity of 81.7%.

## 4. Discussion

To our knowledge, this study provides one of the first evaluations of the potential association between platelet-to-albumin ratio (PAR) and diabetic peripheral neuropathy (DPN) among patients with type 2 diabetes mellitus. We observed that PAR levels were higher in patients with DPN than in those without DPN. Multivariable logistic regression analyses further showed that higher PAR was significantly associated with the prevalence of DPN after adjustment for multiple demographic, metabolic, hematological, and clinical covariates. Restricted cubic spline analyses supported an overall positive linear association between PAR and the prevalence of DPN, while quartile-based analyses suggested that the association became statistically more apparent at higher PAR levels. The quartile-based analysis and spline modeling were complementary rather than contradictory. This may reflect the loss of statistical power after categorizing a continuous variable, such that smaller between-group differences in T2 and T3 did not reach statistical significance. While the spline analysis suggested a generally positive continuous trend between PAR and prevalent DPN, the quartile analysis indicated that the association was more pronounced among individuals with higher PAR levels. However, the magnitude of the association was modest, and ROC analysis showed only limited discriminative ability. The ROC analysis deserves cautious interpretation. Although PAR showed the highest area under the curve among the compared indicators, the overall discriminative performance was modest, with low sensitivity despite relatively high specificity. Therefore, PAR should not be regarded as an independent diagnostic or screening tool for DPN. Its potential clinical relevance lies more in its accessibility and low cost as a complementary laboratory marker that may assist in identifying patients with a more adverse inflammatory–nutritional profile.

DPN is a multifactorial complication of diabetes in which chronic hyperglycemia, insulin resistance, obesity, and dyslipidemia collectively contribute to oxidative stress, microvascular dysfunction, and low-grade systemic inflammation [28]. These metabolic disturbances activate several biological pathways, including the protein kinase C, polyol advanced glycation end-products (AGEs), mechanistic target of rapamycin (mTOR), poly(ADP-ribose) polymerase (PARP), and mitogen-activated protein kinase (MAPK) pathways, ultimately leading to neuronal injury and apoptosis [9,29]. Increasing evidence has suggested that chronic inflammation is closely related to DPN. Several inflammation-related markers, including pro-inflammatory cytokines and hematological indices, have been reported to be associated with DPN in different populations. For example, a longitudinal study demonstrated that higher circulating tumor necrosis factor-α(TNF-α) and intercellular adhesion molecule-1(ICAM-1) were associated with subsequent incident DPN in older patients with T2DM [30]. Other studies have reported associations between hematological inflammatory markers, such as the neutrophil-to-lymphocyte ratio(NLR) or mean platelet volume(MPV), and DPN [31–33]. However, the reported effect sizes vary across studies, likely reflecting differences in study design, population characteristics, and adjustment strategies. The present findings are broadly consistent with the concept that inflammatory and metabolic disturbances are closely linked to DPN.

Consistent with previous studies, patients with DPN in the present cohort exhibited a generally less favorable metabolic and inflammatory profile compared with those without DPN, characterized by poorer glycemic control, dyslipidemia, and higher levels of inflammation-related hematological markers [32–38]. PAR remained associated with DPN prevalence after multivariable adjustment, although the magnitude of the association was modest. The association appeared more apparent at higher PAR levels, whereas weaker associations were observed at lower levels. From a clinical perspective, this modest association is not unexpected. DPN is a complex and heterogeneous complication arising from the cumulative effects of metabolic dysregulation, microvascular injury, oxidative stress, and chronic inflammation over time [39]. A composite biomarker such as PAR is therefore likely to capture only one aspect of this broader pathophysiologic process. This may explain its weak association with DPN when assessed independently. Therefore, these findings suggest that

PAR may provide supplementary information regarding the inflammatory and nutritional milieu of patients with DPN, but its value should not be overstated.

The biological plausibility of PAR is supported by the individual roles of its components. Metabolic abnormalities such as hyperglycemia, insulin resistance, obesity, and dyslipidemia may promote platelet activation, adhesion, and aggregation through inflammatory and oxidative pathways [34,35]. Activated platelets release multiple inflammatory mediators and interact with immune cells, thereby amplifying local inflammatory responses that may contribute to myelin damage and impaired nerve repair [36,37]. In parallel, lower serum albumin levels have been associated with adverse outcomes in several chronic conditions, including cardiovascular disease and diabetic complications. [40,41]. Pro-inflammatory cytokines such as interleukin-1β (IL-1β), interleukin-6 (IL-6), interleukin-8 (IL-8), and TNF-α have been linked to reduced serum albumin concentrations, suggesting that hypoalbuminemia may reflect an underlying inflammatory state [42,43]. In this context, PAR may integrate information from both platelet-related inflammatory activity and albumin-related nutritional and inflammatory status.

Consistent with this interpretation, PAR has been reported to be associated with several inflammation-related conditions, including IgA nephropathy, diabetic nephropathy, cardiovascular diseases, and malignancies. [15,18,27,44]. In the present study, PAR was also associated with several metabolic and inflammatory variables, including glycated hemoglobin, blood pressure, and white blood cell counts. In contrast, it was inversely associated with hemoglobin and albumin levels. Although correlation analyses with C-reactive protein were not performed due to substantial missing data, the observed associations between PAR and hematological inflammatory markers support its role as an indicator of systemic inflammatory-nutritional status in patients with DPN. These findings suggest that PAR reflects a broader metabolic-inflammatory profile relevant to diabetes-related complications. Nevertheless, the present data do not support interpreting PAR as a causal determinant of DPN. Rather, PAR appears to function as an accessible composite marker associated with the clinical and metabolic context in which DPN occurs.

This study has several strengths, including the use of a standardized metabolic management database, comprehensive adjustment for relevant demographic, metabolic, and laboratory factors, and the application of subgroup and spline analyses to examine the consistency of the observed association. Nevertheless, several limitations should be acknowledged. First, the cross-sectional design precludes causal inference, and the observed association between PAR and DPN reflects prevalence rather than disease development over time. Second, although PAR was statistically associated with DPN, the effect size was modest, and its discriminative ability was limited, which restricts its clinical utility as a standalone marker. Additionally, although the MMC network is implemented nationwide, the present analysis was based on data from a single participating center, which may limit the generalizability of the findings. Finally, the underlying biological mechanisms linking PAR to DPN were not directly examined in this study. Prospective and mechanistic studies are needed to further clarify the temporal and biological relationship between PAR and DPN.

## 5. Conclusion

This study found that higher PAR was modestly associated with the prevalence of diabetic peripheral neuropathy in patients with type 2 diabetes mellitus. The platelet-to-albumin ratio (PAR) may reflect underlying inflammatory and nutritional disturbances associated with DPN and should be interpreted within a broader clinical and metabolic context.

## Supporting information

**S1 Fig. Subgroup analyses of the association between platelet-to-albumin ratio (PAR) and diabetic peripheral neuropathy.** The forest plot presents adjusted odds ratios (ORs) with 95% confidence intervals (CIs) for PAR in relation to diabetic peripheral neuropathy across multiple subgroups. Subgroups were defined by age, sex, body mass index, duration of diabetes, history of hypertension, HbA1c, and HDL-C. No significant interactions were observed between PAR and the subgroup variables (all P for interaction > 0.05).
(TIF)

**S1 Table. Receiver operating characteristic (ROC) analysis of PAR and related indicators for diabetic peripheral neuropathy.** Cut-off indicates the optimal threshold value determined by the Youden index. AUC represents the area under the ROC curve, with corresponding 95% confidence intervals (CI). Sensitivity and specificity were calculated at the optimal cut-off point. ALB, serum albumin; PLT, platelet count; PAR, platelet-to-albumin ratio; PHR, platelet-to-high-density lipoprotein cholesterol ratio.
(DOCX)

## Acknowledgments

Thanks to all the colleagues who participated in the study.

## Author contributions

**Conceptualization:** Wenting deng.

**Data curation:** Wenting deng, Qin Wan.

**Formal analysis:** Wenting deng.

**Methodology:** Wenting deng, Siqi Zhang, Yueyang Zhang, Qin Wan.

**Resources:** Siqi Zhang.

**Software:** Wenting deng, Siqi Zhang, Yueyang Zhang, Qin Wan.

**Supervision:** Wenting deng, Yueyang Zhang, Qin Wan.

**Validation:** Wenting deng, Siqi Zhang, Yueyang Zhang.

**Visualization:** Wenting deng, Siqi Zhang, Yueyang Zhang, Qin Wan.

**Writing – original draft:** Wenting deng.

**Writing – review & editing:** Qin Wan.

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
