## [Decision Letter · Decision Letter 0]

23 Dec 2025

PONE-D-25-53697The correlation between Platelet-to-albumin ratio and Diabetic peripheral neuropathy: A cross-sectional study in the Chinese populationPLOS One

Dear Dr. Wan,

Thank you for submitting your manuscript to PLOS ONE. After careful consideration, we feel that it has merit but does not fully meet PLOS ONE’s publication criteria as it currently stands. Therefore, we invite you to submit a revised version of the manuscript that addresses the points raised during the review process.

**ACADEMIC EDITOR:** 

Correlation is a statistical term. Please change with association.The Methods section in the abstract miss important information: when and how the study was conducted, which was the eligible population, which were the inclusion and exclusion criteria etc.The conclusion is too strong for the reported results.Ethics statement is incomplete. Provide the data for ethics approval. Since the study "utilizes a registered dataset from the National Metabolic Management Centre (MMC)" it is unclear how participants gave the informed consent.End the Introduction section with the aim of the study.The methods must be described in such details to allow reproduction/replication. Please find some example for improving this section: "systematically curated by healthcare professionals" how? how many? when (prior to the survey or after?), "Samples that met the specified criteria were extracted from the MMC database from June 2018 and November 2022 and subsequently divided into two groups: DPN and non-DPN." we are in 2025 (almost 2026) please explain the gap. When the data were retrieved? Is the database publicly available? Provide details for VPT test (who applied the test? which training they have? etc.). It sound to be sex not gender. List "other pertinent factors". "patient's height and weight were also recorded" were measured or just recorded? "measured on the right arm using a standard mercury sphygmomanometer" when? (in the morning? in afternoon? etc.); "analysis was adjusted for several factors" how the factors were selected? Why diabetes onset is not aa factor used in adjustement?Check the latest updates of Declaration of Helsinki.Provide the date for ethics approval.In some places, the writing is telegraphic (e.g., "Diagnosis of peripheral neuropathy in type 2 diabetes:(1) 113 Have a clear history of" ; Calculations). In section named Calculations please check for bracketsDefine abbreviations the first time when are used in the body of the manuscript.Statistical analysis seems not to be correctly conducted (a mean of 56.21 and a standard deviation of 50.65 indicate that data does not follow a theoretical normal distribution).It is unclear how duration of diabetes was determined. It is also unclear if the evaluated cohort had or not any type of DM.Correlation analysis seems to have no contribution to the study since logistic regression is reportedLooking to statistical analysis result that the aim of the manuscript is to show the readers the potential of statistical analysis. Please rethink analysis based on clinical reasoning. (think to the clinical relevance of an OR equal to 1.2 when 1 means the same risk in both groups). It is unclear how the adjustments were decided (for example, smoking, alcohol consumption, duration of diabetes proved no statistically significant differences between groups).Do not start a sentence with an abbreviation.Do not duplicate information along the manuscript "The study included 1141 patients with T2DM, who were divided into the DPN group (n=714) and the non-DPN group (n=427)." Although r=0.171, P<0.001 is statistically significant tell nothing from clinical point of view.The discussion is based on statistical significance and it is too strong for the reported results. Your statistical analysis looks like fishing and lack clinical reasoning.Discuss the reported results from clinical point of view."PAR’s moderate predictive value (AUC = 0.602)" this is poor not moderate. "and further multicenter and longitudinal studies are needed to validate this conclusion" As the results looks like, no other studies are neededDiscuss generalizability of the reported results.No pathophysiological explanation of the results results is presented in the Discussion section.Your results do not support the following statement "suggesting PAR as a simple, cost-effective, and accessible biomarker for DPN risk assessment".Please upload the raw data when you submit your revision.

We look forward to receiving your revised manuscript.

Kind regards,

Sorana D. Bolboacă, Ph.D., M.Sc., M.D.

Academic Editor

PLOS One

Journal Requirements:

“The Ministry of Science and Technology of China and Southwest Medical University provided funding for this study through grants 2016YFC0901200 and 2019SQN013.”

3. In the online submission form, you indicated that “The datasets used and/or analyzed during the current study are available from the corresponding author on reasonable request.”

6. If the reviewer comments include a recommendation to cite specific previously published works, please review and evaluate these publications to determine whether they are relevant and should be cited. There is no requirement to cite these works unless the editor has indicated otherwise

Reviewers' comments:

Reviewer's Responses to Questions

**Comments to the Author**

1. Is the manuscript technically sound, and do the data support the conclusions?

Reviewer #1: Yes

Reviewer #2: Yes

2. Has the statistical analysis been performed appropriately and rigorously? 

Reviewer #1: Yes

Reviewer #2: Yes

3. Have the authors made all data underlying the findings in their manuscript fully available?

Reviewer #1: Yes

Reviewer #2: No

4. Is the manuscript presented in an intelligible fashion and written in standard English?

Reviewer #1: Yes

Reviewer #2: Yes

5. Review Comments to the Author

Reviewer #1: Review Comments on "The Correlation between Platelet-to-Albumin Ratio and Diabetic Peripheral Neuropathy: A Cross-Sectional Study in the Chinese Population"

Existing Issues and Improvement Recommendations

1.Improvement and Supplementation of Limitations in Study Design:

Limitations of Cross-Sectional Design in Causal Inference: The current study can only confirm the correlation between PAR and DPN, but cannot clarify the causal relationship (e.g., whether elevated PAR is a cause or a result of DPN). It is recommended to further emphasize this limitation in the discussion section.

2.Optimization of Data Interpretation and Statistical Analysis:

Objective Evaluation of PAR Predictive Efficacy: ROC curve analysis shows that the AUC of PAR is 0.602, with a sensitivity of only 35.9%. Although the specificity reaches 81.7%, the overall predictive efficacy is moderately low, and its clinical utility as a single screening indicator for DPN is limited. It is recommended to objectively analyze this deficiency in the discussion, and propose that PAR be combined with other indicators (such as HbA1c, LDL-C, and other proven DPN risk factors) to construct a prediction model, so as to improve the screening efficacy; at the same time, supplement the possible reasons for the low sensitivity (e.g., the complex pathogenesis of DPN, and a single inflammation-related indicator is difficult to fully cover).

Completeness of Confounder Adjustment: Multivariate regression analysis has adjusted for a number of confounding factors, but it is not clear whether indicators directly related to DPN, such as "history of diabetic foot" and "results of nerve conduction velocity testing", are included (although patients with severe diabetic foot complications are excluded, mild to moderate lesions may still have an impact). It is recommended to supplement the basis for the selection of confounding factors (e.g., based on literature reports, results of univariate analysis, etc.), and verify whether the association between PAR and DPN remains stable after including the above potential confounding factors.

In-depth Exploration of Subgroup Analysis: Subgroup analysis shows that factors such as age, gender, and BMI have no interaction effect, but subgroup stratification analysis based on "PAR quartile grouping" (e.g., differences in the risk of DPN in each subgroup under different PAR levels) has not been conducted. It is recommended to supplement this analysis to further clarify whether the stability of the association between PAR and DPN is affected by PAR levels.

3.Improvement of Details in Methodology and Result Presentation:

Clarification of PAR Calculation Method: The study does not detailedly explain the calculation method of PAR (e.g., the unit of platelet count, the unit of albumin, and whether it is "platelet count (×10⁹/L) / albumin (g/L)"). It is recommended to clarify the specific calculation formula and unit of PAR in the methodology section to avoid misunderstandings by readers.

Verification of the Accuracy of Table Data: There is an abnormal situation in the two groups of data for the "γ-GGT" indicator in Table 1 (45.35 (17.7, 44.9), 49.62 (16.3, 42.9)), where the upper limit of the interquartile range is lower than the median, which may be an input error. It is recommended to verify the accuracy of all table data, correct abnormal values, and ensure data reliability.

Standard Use of Abbreviations: When "PAR", "DPN", and "T2DM" first appear in the abstract, their full names are indicated. However, the full names of some abbreviations (such as "ACR" and "PHR") are not specified when they first appear in the main text. It is recommended to standardize this uniformly and indicate the full names of all abbreviations when they first appear.

Unification of Reference Formats: The formats of some references are inconsistent (e.g., some lack DOI numbers, and the abbreviations of authors' names are not unified). It is recommended to unify and standardize them in accordance with the reference format requirements of PLOS ONE to ensure the accuracy of citation formats.

Clarity of Figures and Supplementary Explanations: The OR values and 95% CIs in Figure 2 (subgroup analysis forest plot) are not clearly presented. It is recommended to optimize the layout of the figure; the ROC curve in Figure 3 does not mark the confidence intervals of each indicator. It is recommended to supplement the 95% CIs of the AUC to fully demonstrate the predictive efficacy.

Reviewer #2: 1- The cross-sectional design precludes causal inference, yet parts of the discussion and conclusions imply predictive or screening utility. Statements suggesting PAR as an “early screening biomarker” should be softened or reframed as risk association rather than prediction. I recommend to emphasize that PAR is associated with prevalent DPN, not predictive of future DPN.

2- The ROC analysis shows an AUC of 0.602, which is generally considered poor discrimination, and sensitivity at the optimal cut-off is only 35.9%.Acknowledge more clearly that PAR alone has limited clinical applicability. Discuss whether PAR adds incremental value beyond established risk factors (such as HbA1c, duration of diabetes, ACR).

3- At multiple points, DPN is reported as 427 cases, while elsewhere it is 714 cases. Carefully verify all sample sizes and ensure consistency across text, tables, and figures.

4- Only the highest quartile (T4) shows a significant association. This threshold effect should be emphasized rather than implying a strong linear risk across all PAR levels. Clarify that clinically meaningful risk appears confined to higher PAR values, supported by quartile and spline analyses.

5- The data availability statement indicates that data are available “upon reasonable request,” which does not fully comply with PLOS ONE data-sharing standards unless adequately justified.

6- missing spaces and inconsistent tense exist in the manuscript. improve language.

6. PLOS authors have the option to publish the peer review history of their article (what does this mean?). If published, this will include your full peer review and any attached files.

Reviewer #1: No

Reviewer #2: No

---

## [Author Response · Author response to Decision Letter 1]

20 Jan 2026

We sincerely thank the Academic Editor and the reviewers for their thorough and constructive comments. We have carefully revised the manuscript in response to all points raised, . All page and line numbers reported below refer to the “Revised Manuscript with Track Changes”, text modifications are marked in red font, unless otherwise specified.

Response to the Academic Editor

We sincerely thank you for your careful evaluation and constructive comments. We have revised the manuscript thoroughly to address every point raised. Below, we provide a detailed point-by-point response, and all changes are reflected in the tracked-changes version.

Comment 1: “Correlation is a statistical term. Please change with association.”

We appreciate this important comment. We agree and have replaced “correlation” with “association” in the title and throughout the manuscript, where we refer to the relationship between PAR and DPN as an epidemiologic association (rather than implying causality or a purely correlational relationship). (see Revised Manuscript with Track Changes, page 1, lines 1; page 2, lines 37;).

Comment 2: “The Methods section in the abstract miss important information: when and how the study was conducted, which was the eligible population, which were the inclusion and exclusion criteria etc.”

Thank you for this helpful comment. We revised the Abstract—Methods to include: (1) study design (cross-sectional), (2) data source (National MMC), (3) study time window (June 2018–December 2023 enrollment), (4) key inclusion/exclusion criteria, and (5) core analytic approach (logistic regression, subgroup analysis, restricted cubic spline, ROC).(Revised Manuscript with Track Changes Abstract—Methods: “This cross-sectional study included adult patients with T2DM enrolled in the national metabolic management center (MMC) database between June 2018 and December 2023. Eligible participants were selected according to predefined inclusion and exclusion criteria and were divided into diabetic peripheral neuropathy (DPN) and non-diabetic peripheral neuropathy (non-DPN) groups. The association between PAR and DPN was evaluated using logistic regression models, with subgroup analyses and restricted cubic spline regression to explore potential effect modification and dose–response relationships. Receiver operating characteristic (ROC) curve analysis was performed to assess the discriminative performance of PAR for DPN prevalence. ”See Revised Manuscript with Track Changes, page 2, lines 28–36).

Comment 3: “The conclusion is too strong for the reported results.”

Thank you for this important comment.We agree and have softened the Conclusion to avoid implying screening/prediction utility. The revised text emphasizes that PAR is associated with the prevalence of DPN and this association may be related to underlying inflammatory processes involved in DPN. (See Revised Manuscript with Track Changes, page 3, lines 62–64; page 29, lines 937–941;).

Comment 4 : “Ethics statement is incomplete. Provide the data for ethics approval. Since the study "utilizes a registered dataset from the National Metabolic Management Centre (MMC)" it is unclear how participants gave the informed consent.” and “Provide the date for ethics approval.”

Thank you for this important comment. We revised the Ethics Approval section to explicitly state that:

• all participants provided written informed consent,

• the study was approved by the ethics committee of the Affiliated Hospital of Southwest Medical University (approval code: 2018017; approval date: February 2018),

• the dataset used for analysis was de-identified prior to statistical analysis.

(Revised Manuscript with Track Changes:

“2.2 Ethics approval

Each participant provided written informed consent related to participation in this study. The study was conducted according to the ethical guidelines of the 2013 2024 Declaration of Helsinki (latest revision) and was approved by the ethics committee of the Affiliated Hospital of Southwest Medical University (ethical approval code: 2018017, date: February 2018). ” See Revised Manuscript with Track Changes, page 6, lines152–156).

Comment 5: “End the Introduction section with the aim of the study.”

Thank you for this helpful comment. We revised the final paragraph of the Introduction to end with a clear, single-sentence aim describing the study objective: “Therefore, this study aimed to investigate the association between platelet-to-albumin ratio (PAR), a composite marker reflecting inflammatory and nutritional status, and the prevalence of diabetic peripheral neuropathy among patients with type 2 diabetes mellitus in a Chinese population.” (See Revised Manuscript with Track Changes, page 5, lines 112–115).

Comment 6: “The methods must be described in such details to allow reproduction/replication. Please find some example for improving this section: "systematically curated by healthcare professionals" how? how many? when (prior to the survey or after?), "Samples that met the specified criteria were extracted from the MMC database from June 2018 and November 2022 and subsequently divided into two groups: DPN and non-DPN." we are in 2025 (almost 2026) please explain the gap. When the data were retrieved? Is the database publicly available? Provide details for VPT test (who applied the test? which training they have? etc.). It sound to be sex not gender. List "other pertinent factors". "patient's height and weight were also recorded" were measured or just recorded? "measured on the right arm using a standard mercury sphygmomanometer" when? (in the morning? in afternoon? etc.)"Analysis was adjusted for several factors." How were the factors selected? Why diabetes onset is not aa factor used in adjustement?”

We sincerely thank you for your careful evaluation and constructive comments. We substantially expanded the Methods to improve reproducibility:

1. Data collection personnel and procedure

Thank you for this comment. We clarified that data was collected by trained endocrinology physicians and nurses, following standardized operating procedures, with uniform training before data collection.(Original text: “The National Metabolic Management Centre 118 database is a national, registered clinical management system with restricted access, and all data used in the present study were de-identified before analysis [22–24]. Data were collected by trained endocrinology physicians and nurses at participating centers following standardized operating procedures, and all personnel received uniform training on study-specific questionnaires and outcome measures before data collection.” See Revised Manuscript with Track Changes, page 5, lines 119–125).

2. Explain the time gap

Thank you for this comment. We clarified that data extraction for the present analysis was conducted in December 2023 using the complete MMC database available at that time. From this database, we retrospectively identified patients enrolled between June 2018 and November 2022 according to predefined inclusion and exclusion criteria. Therefore, the time gap reflects the retrospective nature of data selection rather than delayed data collection. (Original text: “Data extraction was performed in December 2023 from the MMC database. Patients enrolled between June 2018 and November 2022 who met the predefined inclusion criteria were retrospectively identified and subsequently divided into diabetic peripheral neuropathy(DPN) and non-DPN groups for the present analysis.” See Revised Manuscript with Track Changes, page 6, lines 146–150).

3. Database availability

Thank you for this important comment. We clarified that the MMC database is a national, real-world clinical management system with restricted access. As described in previous MMC-related publications, including large multicenter studies, the database contains sensitive clinical and longitudinal health information collected under ethical approval and specific informed consent, which does not permit unrestricted public data sharing. The MMC data governance committee therefore governs data access. In line with established MMC publications, the dataset used in the present study cannot be made publicly available, but de-identified data may be provided upon reasonable request subject to appropriate approval.(see Revised Manuscript with Track Changes, page 5, lines 118–121; page 32, lines 948–956;).

4. VPT/monofilament testing details

Thank you for this comment. We expanded the DPN assessment description and clarified that VPT and monofilament testing were performed by trained endocrinologists or certified technicians who received standardized training and followed uniform procedures across centers.(Original text: “Diagnosis of diabetic peripheral neuropathy was based on standardized clinical criteria. All assessments were performed by trained endocrinologists or certified technicians who had received uniform training in neuropathy evaluation before the study and who followed standardized operating procedures across participating centers. Patients were required to have a confirmed diagnosis of type 2 diabetes mellitus, with neuropathic symptoms occurring at or after diagnosis of diabetes. ”see Revised Manuscript with Track Changes, page 6-7, lines 162–176).

5. Sex vs gender

Thank you for this comment. We replaced “gender” with “sex” throughout the manuscript to ensure correct terminology.[see Revised Manuscript with Track Changes, page 8, lines 361; page 9, lines 478; page 12, lines 544(Table1: Row 2 and Column 1); page 15, lines 525; page 16, lines 536(Table2: Row 5 and Column 1); page 18, lines 551(Table3: Row 3 and Column 1);page 21, lines 561; page 22, lines 566(Table4: Row 9-10 and Column 1);page 22, lines573;page 39, lines 1163].

6. “Other pertinent factors”

Thank you for this comment. We clarified that other sociodemographic variables (e.g., occupation, residence, education) recorded in MMC but were excluded from the present analysis because they were not directly relevant to the study objectives. We have deleted “other pertinent factors”(Original text: “Clinical data were collected as follows: The patient's demographic information includes sex and age. ”See Revised Manuscript with Track Changes, page 8, lines 305-306).

7. “Height and weight recorded”

We appreciate this important comment. We revised wording to clarify that these were measured (not merely abstracted) during the standardized physical examination.(Original text: “The patients' height and weight were also measured. ” See Revised Manuscript with Track Changes, page 8, lines 308).

8. Blood pressure measurement timing

Thank you for this comment. We clarified BP was measured in the morning after ≥5 minutes rest, with three readings averaged. (Original text: “Blood pressure was measured in the morning after at least 5 minutes of rest. ” See Revised Manuscript with Track Changes, page 8, lines 309).

9. Covariate selection and diabetes onset/duration

Thank you for this comment. We clarified that covariates were selected a priori based on clinical relevance, the literature, and univariate analyses, rather than solely on baseline group differences. (Original text: “The analysis was adjusted for several factors, including age, sex, smoking, alcohol consumption, duration of diabetes, SBP, DBP, BMI, FPG, 2hPG, glycated hemoglobin (HbA1c), Hb, RBC, WBC, HCT, MCV, MCH, MPV, ALT, AST, ALP, γ-GGT, BUN, TG, TC, HDL-C, LDL-C, urinary ACR, Cr, and eGFR; covariates included in the multivariable models were selected based on a combination of prior literature, clinical relevance, and univariate analyses. ”See Revised Manuscript with Track Changes, page 10, lines 491–496).

We also explicitly addressed diabetes duration: it was included in the models and was determined from patient-reported time since diagnosis, collected via standardized questionnaires during visits/follow-up. (Original text: “Only participants with confirmed type 2 diabetes mellitus were included in the present analysis. Diabetes duration was determined based on patient-reported time since diagnosis, collected during standardized clinical visits and follow-up. ”see Revised Manuscript with Track Changes, page 6, lines 143–145).

Comment 7: “Check the latest updates of Declaration of Helsinki.”

Thank you for this comment. We updated the ethics phrasing to reflect the manuscript’s stated compliance with the latest revision of the Declaration of Helsinki.(Original text: “The study was conducted according to the ethical guidelines of the 2024 Declaration of Helsinki (latest revision) and was approved by the ethics committee of the Affiliated Hospital of Southwest Medical University (ethical approval code: 2018017, date: February 2018).”See Revised Manuscript with Track Changes, page 6, lines 153–154).

Comment 8: “Provide the date for ethics approval.”

Thank you for this comment. We have provided the ethical date. For further details, please refer to comment 4 and comment 7.

Comment 9: “In some places, the writing is telegraphic (e.g., "Diagnosis of peripheral neuropathy in type 2 diabetes:(1) 113 Have a clear history of"; Calculations). In section named Calculations please check for brackets”.

Thank you for this comment. We revised telegraphic fragments into complete sentences (including the DPN diagnostic criteria section) and corrected formatting issues in the Calculations section (including bracket/format consistency and equation presentation).(Original text: “Diagnosis of diabetic peripheral neuropathy was based on standardized clinical criteria. All assessments were performed by trained endocrinologists or certified technicians who had received uniform training in neuropathy evaluation before the study and who followed standardized operating procedures across participating centers. Patients were required to have a confirmed diagnosis of type 2 diabetes mellitus, with neuropathic symptoms occurring at or after diagnosis of diabetes. Neurological symptoms, including numbness, pain (tingling or prickling, pricking, burning, or aching), and sensory abnormalities (abnormal cold or heat sensations, nociceptive hypersensitivity, and dysesthesia) in the toes, feet, legs, or upper limbs, were assessed by structured questioning. Bilateral Achilles tendon reflexes were examined in the knee standing position and recorded as present, diminished, or absent [26]. The vibration perception threshold (VPT) was assessed at the metatarsophalangeal joint of the hallux using a neurothesiometer (Bio-Thesiometer; Bio-Medical Instrument Co., Newbury, OH, USA). The patients were first instructed on how to recognize the vibration sensation. The stimulus amplitude was gradually increased from zero, and the patients were asked to indicate when the vibration was first perceived. Measurements were performed three times on the plantar surface of each hallux, and the median value was recorded as the VPT for each foot. Sensitivity to touch was evaluated using a 5.07/10-g Semmes-Weinstein monofilament (SWM) at four standardized sites on each foot( three plantar sites and one dorsal side). The monofilament was applied perpendicular to the skin until it just buckled, and was maintained for approximately 2 seconds. Diabetic peripheral neuropathy was defined as the VPT T ≥ 25 V and/or inability to perceive the 10-g monofilament at one or more test sites[27]”See Revised Manuscript with Track Changes, page 6-7, lines 162–303).

Comment 10: “Define abbreviations the first time used in the body”.

Thank you for this comment. We revised the manuscript to define all abbreviations at first occurrence in the main text (including items such as ACR, PHR, etc.).

Comment 11: “Mean 56.21 and SD 50.65 indicates non-normal distribution.”

Thank you for this comment. We clarified that values like “56.21 (50, 65)” are median (IQR), not mean (SD). We revised table formatting and footnotes to clearly present non-normally distributed variables as median (IQR), avoiding misinterpretation.(see Revised Manuscript with Track Changes, page 12, lines 515(Table1).

Comment 12: “It is unclear how duration of diabetes was determined. It is also unclear if the evaluated cohort had or not any type of DM.”

Thank you for this comment. We clarified that diabetes duration was determined based on patient-reported ti

---

## [Decision Letter · Decision Letter 1]

23 Feb 2026

PONE-D-25-53697R1The association between Platelet-to-albumin ratio and Diabetic peripheral neuropathy: A cross-sectional study in the Chinese populationPLOS One

Dear Dr. Wan,

Thank you for submitting your manuscript to PLOS ONE. After careful consideration, we feel that it has merit but does not fully meet PLOS ONE’s publication criteria as it currently stands. Therefore, we invite you to submit a revised version of the manuscript that addresses the points raised during the review process.

We look forward to receiving your revised manuscript.

Kind regards,

Sorana D. Bolboacă, Ph.D., M.Sc., M.D.

Academic Editor

PLOS One

Journal Requirements:

If the reviewer comments include a recommendation to cite specific previously published works, please review and evaluate these publications to determine whether they are relevant and should be cited. There is no requirement to cite these works unless the editor has indicated otherwise

Additional Editor Comments:

Please carefully address all comments and suggestions of the reviewers

Reviewers' comments:

Reviewer's Responses to Questions

**Comments to the Author**

1. If the authors have adequately addressed your comments raised in a previous round of review and you feel that this manuscript is now acceptable for publication, you may indicate that here to bypass the “Comments to the Author” section, enter your conflict of interest statement in the “Confidential to Editor” section, and submit your "Accept" recommendation.

Reviewer #1: All comments have been addressed

Reviewer #3: (No Response)

Reviewer #4: All comments have been addressed

Reviewer #5: All comments have been addressed

2. Is the manuscript technically sound, and do the data support the conclusions?

Reviewer #1: Yes

Reviewer #3: No

Reviewer #4: Yes

Reviewer #5: Yes

3. Has the statistical analysis been performed appropriately and rigorously? 

Reviewer #1: Yes

Reviewer #3: No

Reviewer #4: Yes

Reviewer #5: Yes

4. Have the authors made all data underlying the findings in their manuscript fully available?

Reviewer #1: Yes

Reviewer #3: Yes

Reviewer #4: Yes

Reviewer #5: Yes

5. Is the manuscript presented in an intelligible fashion and written in standard English?

Reviewer #1: Yes

Reviewer #3: No

Reviewer #4: Yes

Reviewer #5: Yes

6. Review Comments to the Author

Reviewer #1: (No Response)

Reviewer #3: I have carefully reviewed the manuscript entitled “The association between Platelet-to-albumin ratio and Diabetic peripheral neuropathy: A cross-sectional study in the Chinese population”. While the topic is clinically relevant, there are several major methodological and reporting issues that prevent acceptance of the manuscript in its current form.

Study design vs. claims:

The study is described as cross-sectional, yet the introduction and discussion suggest predictive or causal interpretations of PAR for diabetic complications. A cross-sectional design does not allow for temporality or risk prediction. Claims regarding the predictive value of PAR must be removed or appropriately moderated.

Registry and sample selection:

The manuscript states that data were derived from the national MMC registry. However, a substantial proportion of patients (967 out of ~3700) did not have diabetes, suggesting that the registry was not specifically designed for T2DM patients.

The total number of patients and geographic coverage do not support the claim of a national registry. This raises serious concerns about the representativeness and generalizability of the sample.

The inclusion/exclusion criteria are inconsistently reported, and there is repetition in the text, which reduces clarity.

Ethical considerations:

The manuscript indicates that “all participants provided written informed consent for this study.” Given that the analysis is based on registry data, it is unclear whether the consent was obtained specifically for this study or as part of enrollment in the registry. This must be clarified.

Outcome measurement and variable definition:

The description of laboratory measurements is unclear. For example, 2-hour plasma glucose (2h PG) is reported to be measured from fasting blood samples, which is inconsistent with standard OGTT procedures.

It is not clear whether patients were assessed only once or followed over time. The timing of laboratory measurements (baseline, most recent, or averaged) must be specified.

Statistical analysis:

The multivariable model includes a very large number of covariates, many of which are highly collinear (e.g., FPG and HbA1c; SBP and DBP; Hb, RBC, HCT; lipid profile variables). There is no report of collinearity diagnostics. This may lead to unstable and uninterpretable regression coefficients.

Spearman correlation is incorrectly used to assess associations between PAR and a binary outcome (DPN). Appropriate methods would include logistic regression or nonparametric comparison tests.

Overall validity and reliability:

Due to the issues outlined above—including unclear registry purpose, inappropriate statistical analyses, ambiguous measurement timing, and overinterpretation of results—the validity and reliability of the findings are seriously compromised.

Conclusion:

Given the cumulative methodological flaws and reporting deficiencies, I recommend rejecting the manuscript in its current form. Substantial revision, including clarification of the registry, ethical considerations, proper statistical analyses, and alignment of claims with the study design, would be required before reconsideration

Reviewer #4: (No Response)

Reviewer #5: This manuscript represents methodologically sound research. The authors made significant changes in the revised version, correctly repositioning PAR as an associated marker rather than a screening tool.

There are some comments to improve the quality of the manuscript.

1. Authors should rightly conclude clinical implications.

2. Authors should address why only T4 shows significance despite "linear" spline.

3. Authors should examine the robustness to different DPN definitions.

7. PLOS authors have the option to publish the peer review history of their article (what does this mean?). If published, this will include your full peer review and any attached files.

Reviewer #1: No

Reviewer #3: **Yes:** Mohammad Aghaali

Reviewer #4: No

Reviewer #5: No

---

## [Author Response · Author response to Decision Letter 2]

15 Mar 2026

Manuscript ID: PONE-D-25-53697R1

Manuscript: The association between Platelet-to-albumin ratio and Diabetic peripheral neuropathy: A cross-sectional study in the Chinese population

Dear Academic Editor and Reviewers,

We sincerely thank you for your careful evaluation of our manuscript and for the constructive comments and suggestions. We have revised the manuscript thoroughly in response to all concerns raised. These revisions have substantially improved the clarity, methodological rigor, and interpretation of the study. All page and line numbers reported below refer to the “Revised Manuscript with Track Changes”, text modifications are marked in red font, unless otherwise specified.

Response to Reviewer #3

Reviewer #3: I have carefully reviewed the manuscript entitled “The association between Platelet-to-albumin ratio and Diabetic peripheral neuropathy: A cross-sectional study in the Chinese population”. While the topic is clinically relevant, there are several major methodological and reporting issues that prevent acceptance of the manuscript in its current form.

Q1: Study design vs. claims:

The study is described as cross-sectional, yet the introduction and discussion suggest predictive or causal interpretations of PAR for diabetic complications. A cross-sectional design does not allow for temporality or risk prediction. Claims regarding the predictive value of PAR must be removed or appropriately moderated.

We thank the reviewer for this important comment. We agree that, because this study is cross-sectional, causal inference and prediction of future risk are not appropriate. Accordingly, we carefully revised the manuscript to remove or moderate language implying predictive or causal interpretations. Specifically, we revised the Abstract, Results, Discussion, and Conclusion to consistently describe PAR as being associated with prevalent DPN, rather than as a predictor or causal factor. We also revised the ROC-related wording to refer to the discriminative ability of PAR for distinguishing patients with and without DPN, rather than its predictive value. In the Discussion and Conclusion, we further clarified that PAR should be interpreted as a complementary associated marker rather than a predictive, causal, or standalone clinical tool. In addition, we strengthened the limitations section by explicitly stating that the cross-sectional design precludes causal inference and does not establish temporality.

Q2: Registry and sample selection:

1) The manuscript states that data were derived from the national MMC registry. However, a substantial proportion of patients (967 out of ~3700) did not have diabetes, suggesting that the registry was not specifically designed for T2DM patients.

The total number of patients and geographic coverage do not support the claim of a national registry. This raises serious concerns about the representativeness and generalizability of the sample.

We sincerely thank the reviewer for this insightful comment regarding the description of the MMC database and the representativeness of our study population. First, the Metabolic Management Center (MMC) is a standardized metabolic disease management network initiated in 2016 by the Chinese Diabetes Society of the Chinese Medical Doctor Association and led by Academician Ning Guang. The MMC model aims to provide standardized diagnosis and management of metabolic diseases such as diabetes through an integrated “one center, one-stop service, one standard” approach. By 2019, more than 700 hospitals across 30 provincial administrative regions in China had established MMC centers, collectively managing over 220,000 patients with metabolic diseases. Importantly, the MMC database includes patients with various metabolic conditions, including diabetes, prediabetes, and individuals undergoing metabolic screening. Therefore, the presence of non-diabetic individuals in the initial dataset is expected. In the present study, only patients with confirmed type 2 diabetes mellitus (T2DM) were included in the final analysis after applying predefined inclusion and exclusion criteria, as described in the Methods section.

Thus, although the MMC network is a nationwide standardized management system, our analysis was conducted using data from a single MMC center rather than the entire national MMC registry. To avoid potential misunderstanding, we have revised the manuscript to clarify this point. Specifically: “The study was a cross-sectional analysis based on data obtained from the National Metabolic Management Centre (MMC) program at the Affiliated Hospital of Southwest Medical University, a participating center in the nationwide MMC network in China.” (See Revised Manuscript with Track Changes, page 5, lines 155–158; page 29, lines 1253–1255).

We have also added a statement in the Discussion section acknowledging that the single-center design may limit the generalizability of the findings. (See Revised Manuscript with Track Changes: “Additionally, although the MMC network is implemented nationwide, the present analysis was based on data from a single participating center, which may limit the generalizability of the findings.” page 25, lines 1139–1141).

3) The inclusion/exclusion criteria are inconsistently reported, and there is repetition in the text, which reduces clarity.

We thank the reviewer for pointing out the lack of clarity in the reporting of inclusion and exclusion criteria. In the revised manuscript, we have streamlined the description to avoid repetition. Specifically, redundant statements such as “Only participants with confirmed T2DM were included” were removed because this criterion had already been defined in the inclusion criteria. In addition, exclusion criteria related to non-diabetic individuals were deleted to avoid logical overlap with the inclusion criteria. The description of participant selection was simplified, and the recruitment period was unified across the manuscript to ensure consistency. These revisions improve the clarity of the study design. (See Revised Manuscript with Track Changes: “Eligible participants were adults (age≥18 years) with confirmed T2DM according to the American Diabetes Association (ADA) "Standards of Care in Diabetes"; Exclusion criteria:1) type 1 diabetes, gestational diabetes and other special types of diabetes; 2)Use of oral antiplatelet and anticoagulant drugs;3)History of infection within 3 months, severe diabetic foot complications, or severe liver or kidney disease; 4)Patients with malignant tumors, hematological disorders; 5)Missing data.” page 6, lines 176–182).

Q3: Ethical considerations:

The manuscript indicates that “all participants provided written informed consent for this study.” Given that the analysis is based on registry data, it is unclear whether the consent was obtained specifically for this study or as part of enrollment in the registry.

We thank the reviewer for this comment. We clarified that written informed consent was obtained at the time of enrollment in the MMC program, and that this consent included permission for the use of anonymized clinical data for research purposes. We revised the Ethics Approval section. (See Revised Manuscript with Track Changes: “All participants provided written informed consent at the time of enrollment in the Metabolic Management Center (MMC) program, which included permission for the use of their anonymized clinical data for research purposes.”, page 6, lines 186–188)

Q4: Outcome measurement and variable definition:

1)The description of laboratory measurements is unclear. For example, 2-hour plasma glucose (2h PG) is reported to be measured from fasting blood samples, which is inconsistent with standard OGTT procedures.

We thank the reviewer for pointing out this ambiguity. In the MMC database, fasting plasma glucose (FPG) and 2-hour plasma glucose (2h PG) are measured using venous blood samples collected during a standard oral glucose tolerance test (OGTT). We have revised the manuscript to clarify this description. (See Revised Manuscript with Track Changes: “FPG and 2h PG were measured using venous blood samples collected during a standard OGTT, with samples obtained after an overnight fast and again 2 hours after glucose ingestion. Additional morning fasting blood samples were collected from each patient after an 8-hour fast.”, page 8, lines 249–252)

2)It is not clear whether patients were assessed only once or followed over time. The timing of laboratory measurements (baseline, most recent, or averaged) must be specified.

We thank the reviewer for this important point. We clarified that although patients enrolled in the MMC program may receive regular follow-up as part of routine clinical management, the present study was designed as a cross-sectional analysis. Accordingly, only measurements obtained at a single time point were used in the present analysis. We added this clarification to the Methods section to avoid misunderstanding regarding temporality or repeated measures use.(See Revised Manuscript with Track Changes: “Patients in the MMC program receive continuous follow-up as part of routine clinical management. For this cross-sectional analysis, only clinical and laboratory data obtained at the first MMC visit were used.”, page 5, lines 160–163)

Q5: Statistical analysis:

The multivariable model includes a very large number of covariates, many of which are highly collinear (e.g., FPG and HbA1c; SBP and DBP; Hb, RBC, HCT; lipid profile variables). There is no report of collinearity diagnostics. This may lead to unstable and uninterpretable regression coefficients. Spearman correlation is incorrectly used to assess associations between PAR and a binary outcome (DPN). Appropriate methods would include logistic regression or nonparametric comparison tests.

We sincerely thank the reviewer for these valuable comments. Following the reviewer’s suggestions, we carefully re-evaluated the statistical analysis and made several revisions to improve the robustness and appropriateness of the methods.

1. Multicollinearity diagnostics

We agree that including multiple correlated clinical variables may lead to multicollinearity and unstable regression coefficients. Therefore, multicollinearity diagnostics were performed prior to constructing the multivariable logistic regression models. Variance inflation factors (VIFs) were calculated for all candidate variables using SPSS.Variables with VIF > 5 were considered to have potential multicollinearity and were excluded from the final model. For example, serum creatinine (Cr) showed multicollinearity with eGFR and was therefore removed. After this screening process, all variables retained in the final model had VIF values below 5, indicating acceptable levels of multicollinearity.The revised multivariable logistic regression results are presented in the updated Table 2 and Table 3(page 16, lines 401; page 18, lines 684;).

2. Removal of Spearman correlation analysis

We appreciate the reviewer’s comment regarding the inappropriate use of Spearman correlation for a binary outcome. Following this suggestion, the Spearman correlation analysis has been removed from the revised manuscript. Associations between PAR and DPN are now evaluated exclusively using logistic regression models (Table 2) and quartile-based analyses (Table 3), which are more appropriate for a binary outcome.

3. Revision of the statistical analysis section

The Statistical Analysis section of the manuscript has been revised to clearly describe:

• the procedure for multicollinearity diagnostics

• the variable selection strategy

• the use of logistic regression models for binary outcomes

These revisions improve the methodological rigor and transparency of the statistical analysis. (See Revised Manuscript with Track Changes: “. Multicollinearity was assessed using variance inflation factors (VIF) derived from a linear regression model including all candidate predictors; redundant covariates with VIF > 5 were excluded. Variables that represent clinical manifestations or consequences of diabetic peripheral neuropathy, such as a history of diabetic foot or nerve conduction study results, were not included as covariates, as they are downstream outcomes of neuropathy rather than independent confounding factors. ”, page 5, lines 160–163).

Response to Reviewer#5

Reviewer #5: This manuscript represents methodologically sound research. The authors made significant changes in the revised version, correctly repositioning PAR as an associated marker rather than a screening tool.

1. Authors should rightly conclude clinical implications.

We thank the reviewer for this important comment. We revised the Discussion and Conclusion to ensure that the clinical implications are stated more appropriately and conservatively. In the revised manuscript, we now emphasize that PAR is an accessible and inexpensive laboratory marker associated with prevalent DPN, but that its clinical utility is limited by the modest effect size and limited discriminative performance observed in our ROC analysis. We explicitly state that PAR should not be interpreted as a standalone screening, diagnostic, or predictive tool. Instead, it may provide complementary information regarding the inflammatory and nutritional status of patients with DPN. (See Revised Manuscript with Track Changes: 4. Discussion; page 5, lines 914–1194)

2. Authors should address why only T4 shows significance despite "linear" spline.

We appreciate this insightful comment. We revised the Discussion to address this point more directly. Specifically, we now explain that the restricted cubic spline (RCS) model evaluates the overall continuous association between PAR and DPN across the full distribution of PAR, whereas the quartile-based analysis compares categorized groups against a reference group. Categorizing a continuous variable into quartiles may reduce statistical power and make smaller differences less likely to reach statistical significance. In our analysis, the odds ratios for T2 and T3 were directionally consistent with a positive association, but only the contrast between T4 and T1 was large enough to reach statistical significance. We therefore clarify that the spline and quartile findings are complementary rather than contradictory. (See Revised Manuscript with Track Changes: “Restricted cubic spline analyses supported an overall positive linear association between PAR and the prevalence of DPN, while quartile-based analyses suggested that the association became statistically more apparent at higher PAR levels. The quartile-based analysis and spline modeling were complementary rather than contradictory. This may reflect the loss of statistical power after categorizing a continuous variable, such that smaller between-group differences in T2 and T3 did not reach statistical significance. While the spline analysis suggested a generally positive continuous trend between PAR and prevalent DPN, the quartile analysis indicated that the association was more pronounced among individuals with higher PAR levels.”; page 21, lines 920–929)

3. Authors should examine the robustness to different DPN definitions.

We thank the reviewer for this thoughtful suggestion. We carefully evaluated the feasibility of conducting sensitivity analyses using alternative definitions of DPN. However, in the MMC database used for the present study, the available related variables were limited to the final conclusion of sensory nerve conduction testing, brachial-ankle pulse wave velocity (baPWV), and ankle-brachial index (ABI). The database did not contain sufficient raw neurological examination components or detailed electrophysiological parameters to reconstruct independent alternative diagnostic definitions of DPN. In addition, baPWV and ABI primarily reflect vascular status rather than neuropathy and therefore were not considered appropriate substitutes for DPN classification. To avoid introducing misclassification or methodological bias, we did not perform sensitivity analyses based on alternative DPN definitions. We have acknowledged this as a limitation in the revised Discussion section.

---

## [Decision Letter · Decision Letter 2]

15 Apr 2026

The association between Platelet-to-albumin ratio and Diabetic peripheral neuropathy: A cross-sectional study in the Chinese population

PONE-D-25-53697R2

Dear Dr. Wan,

We’re pleased to inform you that your manuscript has been judged scientifically suitable for publication and will be formally accepted for publication once it meets all outstanding technical requirements.

Kind regards,

Sorana D. Bolboacă, Ph.D., M.Sc., M.D.

Academic Editor

PLOS One

Additional Editor Comments (optional):

- The data availability statement does not fully comply with PLOS ONE data-sharing standards since no adequate justification was included. Anonimized data should be share to endure reliability of statistical analysis.

- Missing spaces and inconsistent tense exist in the manuscript and must be appropriately corrected before publication.

Reviewers' comments:

Reviewer's Responses to Questions

**Comments to the Author**

1. If the authors have adequately addressed your comments raised in a previous round of review and you feel that this manuscript is now acceptable for publication, you may indicate that here to bypass the “Comments to the Author” section, enter your conflict of interest statement in the “Confidential to Editor” section, and submit your "Accept" recommendation.

Reviewer #5: All comments have been addressed

2. Is the manuscript technically sound, and do the data support the conclusions?

Reviewer #5: Yes

3. Has the statistical analysis been performed appropriately and rigorously? 

Reviewer #5: Yes

4. Have the authors made all data underlying the findings in their manuscript fully available?

Reviewer #5: Yes

5. Is the manuscript presented in an intelligible fashion and written in standard English?

Reviewer #5: Yes

6. Review Comments to the Author

Reviewer #5: The revisions have substantially improved the manuscript's scientific rigor and interpretive accuracy, making it a solid contribution to the diabetic complications literature. It also established groundwork for future prospective studies. I consider this manuscript is well written, discussed with proper citation and proper statistical analysis. Authors made significant efforts to address all the concerns pointed by the reviewer. So, I recommend the publication of this manuscript in the esteemed journal PLOS ONE.

7. PLOS authors have the option to publish the peer review history of their article (what does this mean?). If published, this will include your full peer review and any attached files.

Reviewer #5: No

---

## [Editor Report · Acceptance letter]

PONE-D-25-53697R2

PLOS One

Dear Dr. Wan,

I'm pleased to inform you that your manuscript has been deemed suitable for publication in PLOS One. Congratulations! Your manuscript is now being handed over to our production team.

Kind regards,

on behalf of

Professor Sorana D. Bolboacă

Academic Editor

PLOS One